# Identification and Biological Characteristics of *Mortierella alpina* Associated with Chinese Flowering Cherry (*Cerasus serrulata*) Leaf Blight in China

**DOI:** 10.3390/jof10010050

**Published:** 2024-01-05

**Authors:** Dengke Shao, Yuying Xu, Chunyuan Zhang, Zecheng Lai, Linlin Song, Jiyu Su, Ruixian Yang, Xinhong Jing, Abah Felix, Yakubu Saddeeq Abubakar, Guodong Lu, Wenyu Ye

**Affiliations:** 1China National Engineering Research Center of JUNCAO Technology, Fujian Agriculture and Forestry University, Fuzhou 350002, China; dengkeshao99@163.com (D.S.); chunyuanzhang99@163.com (C.Z.); zecheng_lai@163.com (Z.L.); 2College of JunCao Science and Ecology (College of Carbon Neutrality), Fujian Agriculture and Forestry University, Fuzhou 350002, China; 3Fujian University Key Laboratory for Plant-Microbe Interaction, College of Life Sciences, Fujian Agriculture and Forestry University, Fuzhou 350002, China; xu0yuying@163.com (Y.X.); ay.saddeeq@yahoo.com (Y.S.A.); 4College of Plant Protection, Fujian Agriculture and Forestry University, Fuzhou 350002, China; llsong2332@163.com (L.S.); gzsjy471@163.com (J.S.); fabah11@gmail.com (A.F.); 5School of Environmental Engineering and Chemistry, Luoyang Institute of Science and Technology, Luoyang 471002, China; fairy19790805@163.com; 6Xi’an Greening Management Center, Xi’an 710007, China; jxhongg@163.com; 7Department of Biochemistry, Faculty of Life Sciences, Ahmadu Bello University, Zaria 810281, Nigeria; 8Technology Innovation Center for Monitoring and Restoration Engineering of Ecological Fragile Zone in Southeast China, Ministry of Natural Resources, Fuzhou 350002, China

**Keywords:** *Cerasus serrulata*, pathogenic fungi, *Mortierella alpina*, biological characteristics, pathogenicity

## Abstract

The Chinese flowering cherry (*Cerasus serrulata*), an ornamental tree with established medicinal values, is observed to suffer from leaf blight within Xi’an’s greenbelts. This disease threatens both the plant’s growth and its ornamental appeal. In this study, 26 isolates were obtained from plants with typical leaf blight, and only 3 isolates (XA-10, XA-15, and XA-18) were found to be pathogenic, causing similar symptoms on the leaves of the host plant. Based on sequence alignment, the ITS and LSU sequences of the three selected isolates were consistent, respectively. Following morphological and molecular analyses, the three selected isolates were further identified as *Mortierella alpina.* The three selected isolates exhibited similar morphological characteristics, including wavy colonies with dense, milky-white aerial mycelia on PDA medium. Therefore, isolate XA-10 was used as a representative strain for subsequent experiments. The representative strain XA-10 was found to exhibit optimal growth at a temperature of 30 °C and a pH of 7.0. Host range infection tests further revealed that the representative strain XA-10 could also inflict comparable disease symptoms on both the leaves and fruits of three different Rosaceae species (*Prunus persica*, *Pyrus bretschneideri*, and *Prunus salicina*). This study reveals, for the first time, the causative agent of leaf blight disease affecting the Chinese flowering cherry. This provides a deeper understanding of the biology and etiology of *M. alpina*. This study lays a solid foundation for the sustainable control and management of leaf blight disease in the Chinese flowering cherry.

## 1. Introduction

The Chinese flowering cherry, also known as sakura or cherry blossom, represents an ornamental tree species, blooming beautifully in spring and enjoying global appreciation [1]. The term “Sakura” is typically used to describe deciduous trees belonging to the family Rosaceae [2]. The Chinese flowering cherry holds significant economic value due to its considerable appeal to tourists during its spring bloom. In addition to its aesthetic appeal, it also offers a range of medicinal benefits, including anti-inflammatory properties for the skin, prevention of skin roughness, and suppression of melanin production. It is also used as an ingredient in tea [2,3,4,5]. The polyphenols contained in cherries also aid in improving abnormal serum lipid levels, regulating blood glucose, and reducing obesity [6]. *Cerasus serrulata* (Lindley) Loudon, a member of the Rosaceae family and Cerasus Miller [7,8], serves as an essential ornamental cherry germplasm resource that pervades widely across China, the Korean peninsula, and Japan [9,10,11,12]. Its beautiful flowers, wide adaptability, and high ornamental value contribute to its widespread use in urban and landscape designs [6,13].

With the increased cultivation of *Cerasus* spp., associated diseases have become increasingly prominent [14,15,16,17,18]. However, similar to other arboreal plants, *Cerasus* spp. is not immune to various pathogen infections. Previous research has documented leaf blight disease, caused by *Phytophthora ramorum*, impacting the growth of *Prunus laurocerasus* in Washington state [19]. Furthermore, *Phytophthora cambivora* was reported to have attacked the leaves of *P. campanulata*, causing leaf blight disease in Taiwan [20], and *Fusarium proliferatum* has been discovered causing leaf blight disease in sweet cherry trees in Liaoning province [21]. However, there have been few scientific literature reports concerning leaf blight disease or the associated pathogens in Chinese flowering cherry. Therefore, identification of the pathogens in Chinese flowering cherry is essential for establishing disease management programs.

Currently, an unknown disease causing wilting of the leaves in Chinese flowering cherry was found in Xi’an, China, and was thus named Chinese flowering cherry leaf blight based on its symptoms. This disease initially appears on the margin or tip of the leaves and enlarges consistently. The color of the infected leaves becomes chlorotic in the initial stage, and later gradually turns into a brown shade. In the later stages of the disease, a significant number of leaves become infected, inhibiting a tree’s growth. The horticultural value of *C. serrulata* in China and the presence of leaf blight stimulated our interest in understanding which pathogen causes Chinese flowering cherry leaf blight. Therefore, the aim of this study was to detect and identify the pathogen associated with leaf blight of Chinese flowering cherry and to investigate the biological characteristics and pathogenicity of this pathogen on other Rosaceae plants. This study provides essential information to better control the disease and further the sustainable development of the Chinese flowering cherry industry.

## 2. Materials and Methods

### 2.1. Disease Investigation and Isolate Collection

From 2019 to 2022, a field survey of Chinese flowering cherry leaf blight was conducted in the urban street greenbelt of Xi’an (elevation: 400 m; geographic position: 107°40′–109°49′ E, 33°42′–34°45′ N), Shaanxi province, China. A total of 1140 trees were surveyed, and the incidence of leaf blight was approximately 15% from the Xi’an Greening Management Center. To isolate the pathogens, 150 leaves with leaf blight symptoms were collected from 285 trees in three greenbelts. Among them, 30 leaves showing typical leaf blight symptoms and without other disease lesions were randomly selected for pathogen isolation. The junction between the diseased and healthy parts of the typical leaf blight samples was used as experimental material. 

The tissue isolation method was utilized to isolate the pathogenic fungi [22]. Briefly, the leaves with lesions were cleaned with sterile water, the moisture was absorbed, and 2 cm^2^ of the disease area was excised. The samples were then surface sterilized in 75% ethanol (Shandong Anweishi Medical Technology Co., Ltd., Dezhou, China) for 30 s, soaked in 2.0% NaClO (Sinopharm Chemical Reagent Co., Ltd., Shanghai, China) for 2 min, rinsed 3 times with sterile water, and dried with sterile filter paper. Subsequently, the diseased leaves were placed onto potato dextrose agar (PDA) medium (prepared by adding 46 g PDA powder to 1 L of sterile double-distilled water) provided by Guangdong Huankai Microbial and Tech. Co., Ltd., Guangzhou, China, and were incubated at 28 °C in the dark until mycelia appeared. The mycelia surrounding the disinfected leaves were selected for further purification; pure cultures were obtained after more than 3 successive subcultures and then stored at 4 °C for further study.

### 2.2. Pathogenicity Tests on Host Leaves

All isolates were tested for pathogenicity on wounded leaves of the host plant (Chinese flowering cherry). The leaves used in the experiment were sourced from Fujian Agriculture and Forestry University. The healthy samples were cleaned with sterile water, sterilized with 75% ethanol for 30 s, soaked in 2.0% NaClO for 2 min, rinsed with sterile water more than 3 times, and then artificially wounded using sterile toothpicks. Mycelial discs (5 mm in diameter), from the edges of 7-day-old isolated fungal colonies, were placed on the surface of the detached leaf samples, ensuring the mycelia directly contacted the plant tissues. Leaves inoculated with sterile water and PDA without fungal mycelium were used as controls, respectively. To maintain high relative humidity, all inoculated leaves were placed in the culture dish with wet filter paper and then incubated at 28 °C for 5 days.

Meanwhile, mycelial discs (5 mm diameter) were also used to infect the leaves of live host plants at a temperature range of 28–30 °C. All the inoculated samples underwent daily observation, and symptoms were recorded and photographed. All experimental treatments were performed in triplicate. In addition, the pathogens were re-isolated from the leaves of the infected host, their morphological characteristics were analyzed, and the ITS and LSU sequences were amplified for sequence alignment to fulfill Koch’s postulates.

### 2.3. Molecular Identification of Fungal Isolates

Molecular identification of the 3 selected isolates was performed using the internal transcribed spacer (ITS) and large ribosomal subunit (LSU) gene sequences analysis methods with PCR (polymerase chain reaction) [23]. The isolates were cultured on PDA medium at 28 °C for 7 days. Mycelia were harvested from the cultures, and genomic DNA (gDNA) was extracted using the CTAB method [24]. The gDNA samples were then utilized as templates to amplify the ITS and LSU gene sequences, respectively. All amplified loci, primers, and PCR conditions are listed in Table 1. The TIANGEN Golden Easy PCR kit (TIANGEN Biotech, Beijing, China) was used to carry out the PCR. The PCR amplification reaction was adapted from Liu et al. [25] and Yan et al. [26]. The PCR products with evident bands, detected with 1% agarose gel electrophoresis, were sent to Tsingke Biotech, Fuzhou, China, for sequencing. The sequences were confirmed with a BLAST (Basic Local Alignment Search Tool) search of the NCBI (National Center for Biotechnology Information) database (https://www.ncbi.nlm.nih.gov/, accessed on 16 November 2023). The strains utilized in this study and their corresponding GenBank accession numbers are listed in Table 2.

### 2.4. Phylogenetic Analysis

A phylogram was constructed using PhyloSuite v.1.2.1 [33] software (https://github.com/dongzhang0725/PhyloSuite, accessed on 16 November 2023). The construction of the phylogram for the 3 selected isolates, which was based on the combined sequences of two genes (ITS and LSU), was carried out using maximum likelihood (ML) and Bayesian inference (BI) [34]. IQ-TREE v.1.5.5 [35] was used for ML analysis, with the selection of GTR+G (ITS) and TRN+I+G (LSU) models, and the estimation of branch stability was performed with 1000 bootstrap replicates. BI analysis was performed in PhyloSuite v.1.2.1 using Mr Bayes v3.2.6 [36] under a partition model (a total of 2 × 10^7^ generations were run, with trees sampled once every 100 generations). FigTree v.1.4.4 was used to display the phylogram.

### 2.5. Morphological Study

For the colony morphology, the 3 selected isolates were inoculated on PDA medium and incubated at 28 °C in the dark. After 7 days, the colony morphologies from both front and back were analyzed. To investigate the formation of sporangiospores, the representative strain was inoculated on oatmeal agar (OA) medium (50 g oatmeal, and 15 g agar in 1 L of sterile double-distilled water; Guangdong Huankai Microbial and Tech. Co., Ltd., Guangzhou, China) and incubated at 28 °C in the dark. After 7 days, sporangiospores were harvested, observed, and photographed with an Olympus-BX53F (Olympus Corporation Co., Ltd., Tokyo, Japan) light microscope. The representative strain was identified physically by colony characteristics, hyphal structures, and sporangiospores [37].

### 2.6. Growth Characteristics at Different Temperatures

The optimum temperature for the representative strain was evaluated. The representative strain on the PDA medium was incubated under varying temperatures of 15 °C, 20 °C, 25 °C, 30 °C, and 35 °C in the dark. For sporulation, the representative strain was inoculated on OA medium under the same conditions. After 5 days, the colony morphologies were observed and photographed, and the colony diameters of the resulting were measured using vernier calipers (Dongguan Sanliang Measuring Tools Co., Ltd., Dongguan, China). After 7 days of incubation, the sporangiospores were harvested and the sporulation ability was determined.

### 2.7. Growth Characteristics at Different pH Values

The optimum pH of the representative strain was assessed. The growth medium pH was adjusted using 1.0 mol/L HCl or NaOH (Sinopharm Chemical Reagent Co., Ltd., Shanghai, China). The representative strain was inoculated on PDA and OA medium with pH values of 5.0, 6.0, 7.0, 8.0, 9.0, and 10.0 for the growth assay and sporulation, respectively, and then incubated in the dark at 28 °C. After 5 days, the colony diameters were gauged using vernier calipers, and the colony morphologies were observed and photographed. After 7 days of incubation, the sporangiospores were collected for an assessment of their sporulation ability.

### 2.8. Pathogenicity Tests on Leaves and Fruits of the Other Three Species of Rosaceae

To test the host range of the representative strain, inoculation experiments were carried out on the leaves and fruits of other Rosaceae plants, including *Prunus persica*, *Pyrus bretschneideri*, and *Prunus salicina*. The leaves and fruits of the plants used in the experiment were sourced from Fujian Agriculture and Forestry University. Pathogenicity tests using the leaves and fruits of the three Rosaceae species were carried out as previously described in Section 2.2. The inoculated leaves and fruits were then placed in dishes with wet filter papers to maintain a moist microenvironment and incubated at 28 °C for 5 days. All inoculated samples were observed daily, and symptoms were recorded and photographed. All treatments were performed in triplicates.

### 2.9. Statistical Analysis

Statistical analysis and graphing were conducted using Graph Pad Prism version 8.0 (Graph Pad Software, San Diego, CA, USA). One-way analysis of variance (ANOVA) and the least significant difference (LSD) were utilized to evaluate the levels of significance among the samples. All experiments were conducted in triplicate.

## 3. Results

### 3.1. Natural Symptoms and Pathogen Isolation

The initial symptoms on *C. serrulata* leaves were small chlorotic lesions present on the leaf margin or tip. These lesions eventually enlarged to form larger brown lesions, resulting in leaf wilting and severely impeding the plant’s growth (Figure 1A). The results of the isolates showed that a total of 26 single hyphae fungal isolates, coded as XA-1 to XA-26, were obtained from the leaves with typical symptoms of *C. serrulata*.

### 3.2. Pathogenicity Tests on Host Leaves

The pathogenicity of these isolates was tested, revealing discernible differences among the 26 isolates studied (Appendix A). Among them, three isolates (XA-10, XA-15, and XA-18) were found to be pathogenic, exhibiting similar disease symptoms on the host leaves (Figure 1B).

Symptoms manifested on the leaves after 5 days post-inoculation. The detached leaves inoculated with the three selected isolates (XA-10, XA-15, and XA-18) initially showed small brown circular spots (Figure 1B). With increasing inoculation time, the spots gradually expanded, causing the surrounding tissues to turn yellow, with some leaves showing signs of wilting (Figure 1B(c–e)). Similarly, the respective inoculated sites on the leaves of the infected live host plants showed circular spots, and the lesion areas were dark brown. The lesions spread, and the central and inoculated sections of the leaves gradually wilted (Figure 1B(h–j)). Pronounced disease symptoms were identifiable on both the detached leaves as well as the leaves of live host plants. Leaves subjected to inoculation with sterile water (Figure 1B(a,f)) and those inoculated with PDA medium devoid of mycelia (Figure 1B(b,g)) exhibited no symptoms.

To confirm Koch’s postulates, the pathogens were successfully re-isolated from the infected leaves, and their morphological characteristics and gene sequence matched those of the original isolates.

### 3.3. Molecular Identification of Fungal Isolates

The internal transcribed spacer (ITS) and large ribosomal subunit (LSU) regions of isolates XA-10, XA-15, and XA-18, exhibiting similar pathogenicity, were amplified and sequenced. The ITS and LSU sequences were submitted to the GenBank database under the accession numbers ITS: OR206530, OR817763, and OR817762; and LSU: OR206531, OR817765, and OR817764, respectively. The phylogram of the ITS and LSU gene sequence combinations was constructed using maximum likelihood (ML) and Bayesian inference (BI). The topological structures in the two phylograms were similar, and finally, the maximum likelihood phylogram was selected for presentation (Figure 2). Following sequence alignment, the ITS and LSU sequences of the three selected isolates (XA-10, XA-15, and XA-18), respectively, exhibited consistency. As shown in Figure 2, isolates XA-10, XA-15, and XA-18 (accession no. ITS: OR206530, OR817763, and OR817762; accession no. LSU: OR206531, OR817765, and OR817764) and *Mortierella alpina* (GenBank accession number: CBS 250.53) were present in the same cluster with a bootstrap support value of 96% and a posterior probability of 0.99. Based on ITS and LSU sequence analysis, the isolates XA-10, XA-15, and XA-18 were identified as *M. alpina* (Figure 2).

### 3.4. Morphology of Pathogenic Fungi

The three selected isolates (XA-10, XA-15, and XA-18) display similar milky-white aerial mycelia with wavy colony edges when observed from both the front and the back sides of their PDA medium (Figure 3A–C). The mycelium of the representative strain XA-10 (based on the similar morphological characteristics, pathogenicity to host plants, and molecular biological identification of the isolates XA-10, XA-15, and XA-18, the strain XA-10 was selected as the representative sample for the further analysis) exhibits multiple septa and branches (Figure 3D). The fungal sporangia appear to be spherical with smooth surfaces (Figure 3E). The sporangia have an average length and width of 10.09 ± 2.02 μm and 8.94 ± 1.92 μm, respectively. The sporangiospores appear spherical or nearly spherical (Figure 3F). The average length of the sporangiospores is 3.90 ± 0.8 μm, and the average width is 3.76 ± 0.77 μm.

### 3.5. Growth Characteristics at Different Temperatures

The growth and sporulation capabilities of isolate XA-10 were assessed at various temperature conditions (Figure 4). After 5 days of culture on PDA medium at temperatures ranging from 15 to 35 °C, the mycelial growth was significantly (*p* < 0.05) different among different temperatures (Figure 4A–E). The colony diameters of the isolate XA-10 ranged from 0.18 ± 0.03 cm to 6.40 ± 0.04 cm. The maximum and minimum colony diameters were recorded at 30 °C and 35 °C, respectively (Figure 4F). Optimum sporulation was observed at a temperature of 30 °C (1.40 × 10^6^ spores/mL), while the lowest (3.17 × 10^5^ spores/mL) was recorded at 35 °C (Figure 4G). The results indicate that the optimal temperature for the growth and development of isolate XA-10 was 30 °C (Figure 4), with growth nearly ceasing at temperatures exceeding 35 °C (Figure 4D,E). Thus, temperature could significantly impact the sporulation ability of isolate XA-10.

### 3.6. Growth Characteristics at Different pH Values

The effects of varying pH values on the mycelial growth and sporulation capacity of the isolate XA-10 are shown in Figure 5. The isolate showed similar growth characteristics within the pH value range of 5.0–10.0 (Figure 5A–F), and the colony diameters ranged from 5.53 ± 0.04 cm to 6.46 ± 0.03 cm (Figure 5G). The mycelia grew well at pH values between 5.0 and 10.0, and a pH range of 5.0 to 9.0 was more suitable for mycelial growth (Figure 5A–E). Sporulation was highest (1.47 × 10^6^ spores/mL) at neutral pH (7.0) (Figure 5H). The lowest sporulation was observed at pH 10.0 (6.83 × 10^5^ spores/mL), as shown in Figure 5H. Although no significant differences were found in mycelial growth under various pH conditions (*p* < 0.05), the sporulation of isolate XA-10 was significantly affected by pH.

### 3.7. Pathogenicity Tests on Leaves and Fruits of the Other Three Species of Rosaceae

The leaves and fruits of *P. persica*, *P. bretschneideri*, and *P. salicina* inoculated with isolate XA-10 showed severe symptoms, with an incidence of 100%, excluding the asymptomatic control (Figure 6). 

After 2 days of inoculation, brown lesions emerged on the leaves and fruits of *P. persica*. These infected regions noticeably expanded after 4 days, causing extensive wilting of the leaves (Figure 6(A3)) and severe rotting of the pulp, producing a foul odor (Figure 6(A6)). On *P. bretschneideri* and *P. salicina*, however, black and brown lesions were observed 3 days post-inoculation. The symptoms on the leaves of *P. bretschneideri* continued to expand, displaying irregular shapes after 5 days (Figure 6(B3)). At this time, the infected fruits had significantly rotted internally, with the initially small, round, black lesions at the inoculation site gradually spreading (Figure 6(B6)). Similarly, brown lesion symptoms were observed on the leaves of *P. salicina*, and the distribution of these spots showed an irregular pattern (Figure 6(C3)). The fruits of *P. salicina* showed significant rot accompanied by a foul odor and disease spots (Figure 6(C6)). Thus, isolate XA-10 is pathogenic to *P. persica*, *P. bretschneideri*, and *P. salicina*, causing similar symptoms on the leaves and fruits following wound inoculation. The control group displayed no symptoms (Figure 6).

## 4. Discussion

The Chinese flowering cherry, renowned for its vibrant colors and 15-20-day flowering period, is a popular choice for urban green spaces and parks due to its significant ornamental value [38]. Leaf blight detrimentally impacts not only the aesthetics of plants but also their growth and development. In severe cases, it leads to the death of the plant, threatening China’s economic development and ecological environment [19,20,21]. In our study, the fungal pathogen causing leaf blight was isolated from a diseased plant, identified by morphological and cultural characteristics [39], and confirmed to be *M. alpina* using ITS [23] and LSU [27,28] sequence analyses. This report is the first to implicate *M. alpina* as the causative agent of leaf blight in the Chinese flowering cherry. Consequently, further exploration is required to understand the role of this fungus as a leaf blight pathogen that affects economically significant plant species.

*M. alpina* is a member of the fungus Zygomycotina [40]. The morphological traits of the pathogen described in this study, particularly the colony morphology and milky-white aerial mycelia, were consistent with the species description of *M. alpina* previously established by Wani et al. [41] and Wang et al. [42]. Other features, including the sporangia’s shape and the sporangiospores, correspond to those characterized by Yadav et al. [43] for *M. alpina*.

Environmental factors, such as temperature and pH, were the key factors influencing fungal mycelia’s growth. The peak temperature for mycelial growth in this study was similar to that of Streekstra et al. [44] and Zhang [45]. However, the optimal temperature for the growth of fungal mycelia differed slightly from that reported by Melo et al. [46]. These variations in biological characteristics may stem from the different hosts and regions evaluated in these studies. Most phytopathogenic fungi grow optimally within a pH range of 5.0 to 6.5 [47]. However, we found that *M. alpina* exhibited optimal mycelial growth at pH 7.0, which was consistent with the results of Zhang [45].

Results from some previous studies suggest that some members of the genus *Mortierella* are weak or conditioned pathogens, as well as rhizosphere microorganisms or endophytes of plants [41,46,48,49]. A common member of this genus, *M. alpina*, can induce several plant diseases, such as seedling necrosis of *Pinus nigra* [50] and seedling necrosis of the phloem of *Araucaria araucana* [51]. *M. elongata* has been found to cause diseases in avocado plants [52]. Furthermore, our study observed that *M. alpina* can severely infect the leaves and fruits of three other Rosaceae plants, including *P. persica*, *P. bretschneideri*, and *P. salicina*. Yet, the mechanism by which *Mortierella* spp. instigate plant diseases remains unidentified. In future studies, we aim to unravel the pathogenic mechanisms of these fungal isolates causing leaf blight in the Chinese flowering cherry.

*Mortierella* spp., acting as rhizospheric or endophytic fungi, plays essential roles in the plant–soil ecosystem, plant growth, and health [41,48,53,54,55]. Generally, *Mortierella* spp., being filamentous fungi, are commonly found in soil, the rhizosphere, and plant tissues [53,54,56,57,58]. Previous research has indicated that *Mortierella* species are present in the rhizosphere soil of *Panax notoginseng* and a poplar plantation, suggesting that they may play an important role in maintaining plant–soil balance or assisting host plants in the uptake of phosphorus and nitrogen [53,57]. *Mortierella* spp. may be an important component of the phosphorus cycle [54]. Certain *Mortierella* species have the capability to convert insoluble phosphorus into a soluble form, thereby promoting plant growth and development [56]. *Mortierella* species have been demonstrated to positively impact wheat yield [59], with *M. elongata* found abundantly in the rhizosphere of pea farms preceded by wheat in crop rotation [60]. *M. capitata* could increase maize biomass and promote plant growth [61]. Many *Mortierella* species can also improve soil nutrition, reconstruct microbial communities, and improve litter decomposition in the rhizosphere [56,61,62,63].

Further investigation is warranted into the prevalence of *M. alpina* infection among diverse horticultural plants. This study provided a theoretical foundation and practical guide for managing leaf blight disease in Chinese flowering cherry, among other Rosaceae plants. Nonetheless, additional studies are needed to elucidate the host range and the mechanisms underlying the pathogenicity of this fungus.

## 5. Conclusions

In summary, this study identified the pathogen causing Chinese flowering cherry leaf blight disease as *M. alpina* based on morphological observations, molecular identification, and pathogenicity testing. To the best of our knowledge, this is the first report of Chinese flowering cherry leaf blight disease caused by *M. alpina* in China. Its biological characteristics are as follows: the strain XA-10 is suitable for mycelial growth at 30 °C, and the optimal pH value is 7.0 for mycelial growth. This fungus causes varying degrees of symptoms on leaves and fruits of other Rosaceae plants including *P. persica*, *P. bretschneideri*, and *P. salicina*. Our results enrich the knowledge about the specific features of *M. alpina* and serve as important scientific foundations for efficient control and management of the studied plant disease.

## Figures and Tables

**Figure 1 jof-10-00050-f001:**
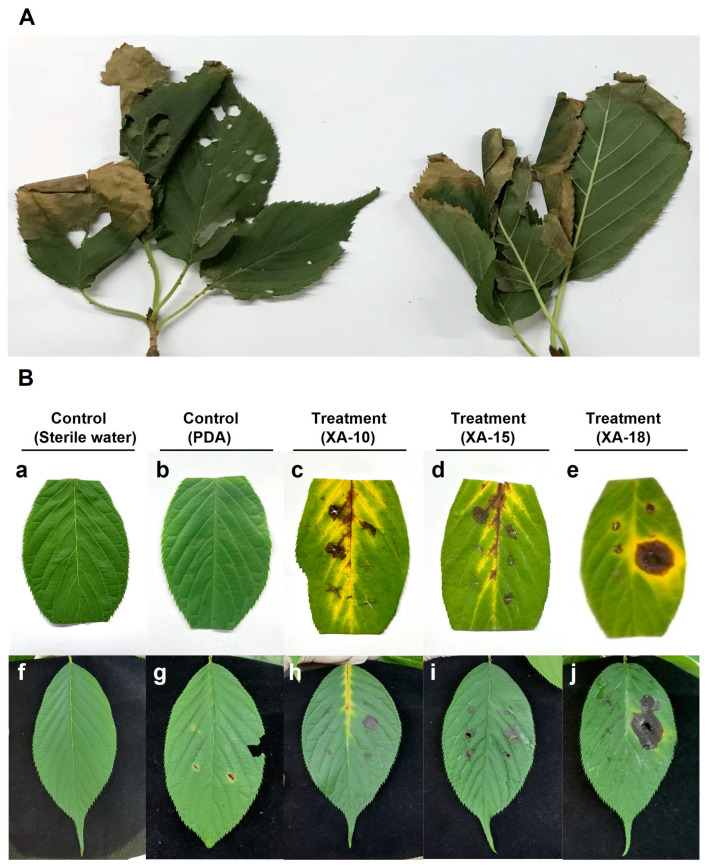
Disease symptoms in greenbelts and pathogenicity tests of the 3 selected isolates (XA-10, XA-15, and XA-18) on host leaves. (**A**) Symptoms on naturally infected leaves of *Cerasus serrulata*. (**B**) Pathogenicity test for leaf blight symptoms caused by mycelial inoculation of the 3 selected isolates on the host plant (flowering cherry) leaves after 5 days. (**B**: **a**–**e**) Pathogenicity on detached leaves of host plants after 5 days of inoculation. (**B**: **f**–**j**) Pathogenicity on leaves of live host plants after 5 days of inoculation.

**Figure 2 jof-10-00050-f002:**
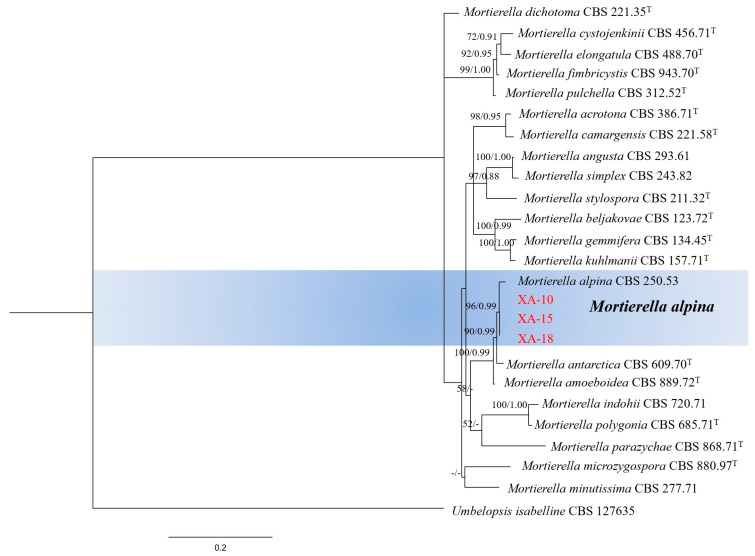
Maximum likelihood (ML) phylogram of the 3 selected isolates (XA-10, XA-15, and XA-18) based on the combined 2-locus (ITS and LSU) sequences. The tree is rooted in *Umbelopsis isabellina* (CBS 127635). Maximum likelihood bootstrap values ≥ 50% and Bayesian posterior probabilities ≥ 0.85 (PP/MLBS) are given at the nodes. Maximum likelihood bootstrap values < 50%, Bayesian posterior probabilities < 0.85 (PP/MLBS) are indicated with “-”. The type species is represented by “^T^”. The 3 selected isolates in this study are shown in red. The scale bar represents the expected number of changes per site.

**Figure 3 jof-10-00050-f003:**
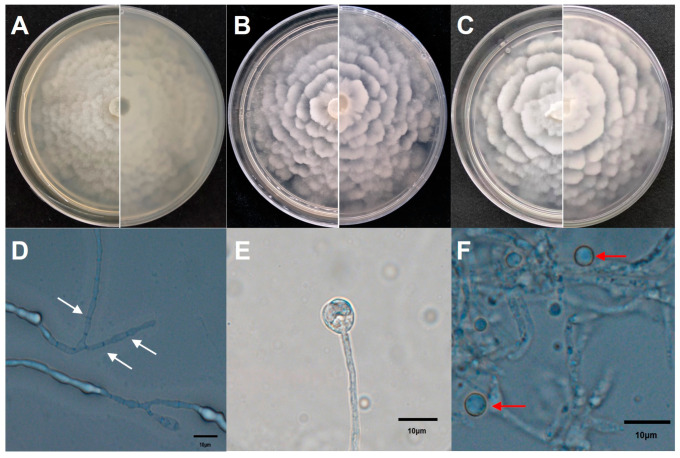
Morphological features of *Mortierella alpina*. (**A**–**C**) Colony morphologies (front and back) of the 3 selected isolates (XA-10, XA-15, and XA-18) on PDA medium incubated at 28 °C for 7 days. (**D**–**F**) Septate hyphae (white arrows), sporangium, and spherical and nearly spherical sporangiospores (red arrows) of the representative strain XA-10 after 7 days of incubation on OA medium at 28 °C. Scale bar = 10 µm.

**Figure 4 jof-10-00050-f004:**
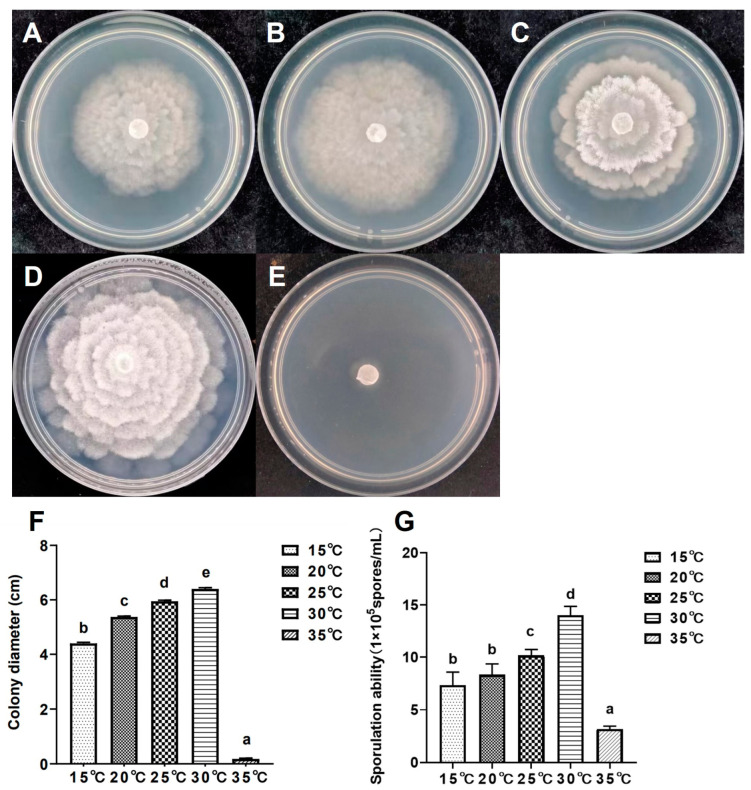
Colony morphology, diameter, and sporulation ability of the representative strain XA-10 at different temperatures. (**A**–**E**) Colony morphology of isolate XA-10 at 15–35 °C on PDA medium. (**F**,**G**) Colony diameter and sporulation quantity of isolate XA-10 at different temperatures. Values of ±standard error are represented with error bars, and different letters indicate significant differences at *p* < 0.05.

**Figure 5 jof-10-00050-f005:**
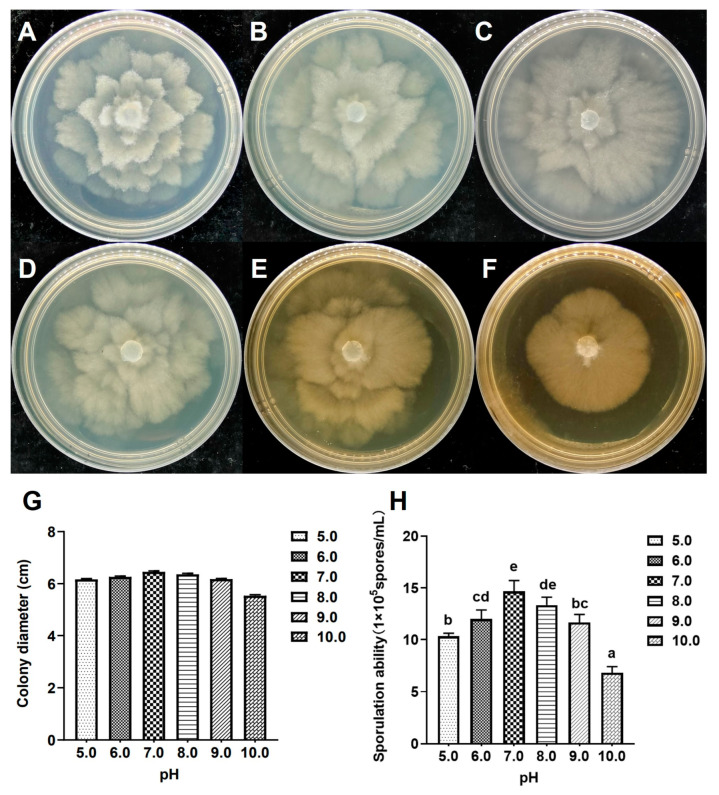
Colony morphology, diameter, and sporulation ability of the representative strain XA-10 at different pH values. (**A**–**F**) Colony morphologies of isolate XA-10 at pH 5.0–10.0 on PDA medium. (**G**,**H**) Colony diameters and sporulation ability of isolate XA-10 at different pH values. Values of ± standard error are represented with error bars, and different letters indicate significant differences at *p* < 0.05.

**Figure 6 jof-10-00050-f006:**
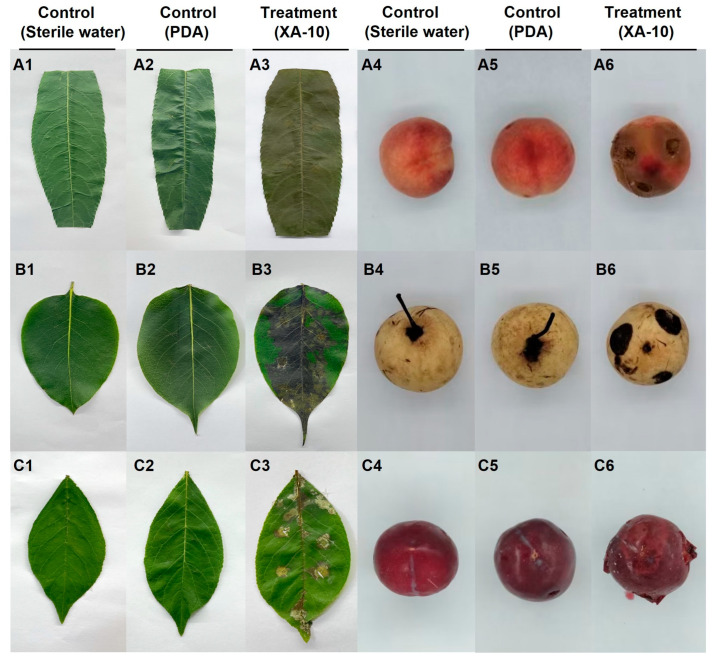
Pathogenicity of the representative strain XA-10 on leaves and fruits of three other Rosaceae plants. (**A1**–**A6**) Detached leaves and fruits of *Prunus persica*; (**B1**–**B6**) detached leaves and fruits of *Pyrus bretschneideri*; and (**C1**–**C6**) detached leaves and fruits of *Prunus salicina*. The photographs of *P. persica* were taken 4 days post-inoculation and of *P. bretschneideri* and *P. salicina* were taken 5 days post-inoculated.

**Table 1 jof-10-00050-t001:** Amplification sites, primer sequences, and PCR conditions used in this study.

Locus ^a^	Primer	Primer Sequence (5′–3′)	PCR Conditions	Reference
ITS	ITS5	GGAAGTAAACGTAACAAGG	94 °C for 5 min (94 °C for 40 s, 58 °C for 40 s, and 72 °C for 60 s) × 35 cycles, 72 °C for 7 min	[23]
	ITS4	TCCTCCGCTTATTGATATGC		
LSU	LR0R	ACCCGCTGAACTTAAGC	94 °C for 5 min (94 °C for 45 s, 50 °C for 45 s, and 72 °C for 1 min) × 35 cycles, 72 °C for 7 min	[27,28]
	LR5	TCCTGAGGGAAACTTCG		

^a^ Genes: ITS, internal transcribed spacer; LSU, large ribosomal subunit.

**Table 2 jof-10-00050-t002:** Strains used in this study.

Original Name	Culture Accession Number(s)	Type Status	Accession No. ITS	Accession No. LSU	Reference
*Mortierella acrotona*	CBS 386.71, FSU 9788	Type of *Mortierella acrotona*	JX975921	HQ667405.1	[29,30]
*Mortierella alpina*	CBS 250.53		JX975955	KC018184	[29]
*Mortierella amoeboidea*	CBS 889.72	Type of *Mortierella amoeboidea*	JX976073	HQ667422.1	[29,30]
*Mortierella angusta*	CBS 293.61	Neotype of *Mortierella polycephala* var.*angusta*	JX976061	HQ667358.1	[29,30]
*Mortierella antarctica*	CBS 609.70, FSU 9792	Type of *Mortierella antarctica*	JX975907	HQ667503.1	[29,30]
*Mortierella beljakovae*	CBS 123.72, FSU 9794	Type of *Mortierella beljakovae*	JX976126	HQ667428.1	[29,30]
*Mortierella camargensis*	CBS 221.58, FSU 9796	Type of *Mortierella camargensis*	JX975949	HQ667408.1	[29,30,31]
*Mortierella cystojenkinii*	CBS 456.71, FSU 9803	Type of *Mortierella cystojenkinii*	JX976030	HQ667504.1	[29,30]
*Mortierella dichotoma*	CBS 221.35, FSU 9804	Syntype of *Mortierella dichotoma*	JX975842	HQ667393.1	[29,30]
*Mortierella elongatula*	CBS 488.70, FSU 9808	Type of *Mortierella elongatula*	JX975967	HQ667425.1	[29,30]
*Mortierella fimbricystis*	CBS 943.70	Type of *Mortierella fimbricystis*	GU559986.1	JX976172	[29]
*Mortierella gemmifera*	CBS 134.45, FSU 9815	Type of *Mortierella gemmifera*	JX975931	HQ667371.1	[29,30]
*Mortierella indohii*	CBS 720.71, FSU 9826	Isotype of *Mortierella indohii*	JX975856	HQ667377.1	[29,30]
*Mortierella kuhlmanii*	CBS 157.71, FSU 9827	Type of *Mortierella kuhlmanii*	JX975846	HQ667372.1	[29,30]
*Mortierella microzygospora*	CBS 880.97, FSU 9831	Type of *Mortierella microzygospora*	JX976027	HQ667394.1	[29,30]
*Mortierella minutissima*	CBS 277.71, FSU 832		JX975938	KC018293	[29]
*Mortierella parazychae*	CBS 868.71, FSU 9836	Type of *Mortierella parazychae*	JX975985	HQ667362.1	[29,30]
*Mortierella polygonia*	CBS 685.71, FSU 9839	Type of *Mortierella polygonia*	JX975900	HQ667378.1	[29,30]
*Mortierella pulchella*	CBS 312.52, FSU 9840	Authentic strain of *Mortierella pulchella*	JX976054	HQ667427.1	[29,30]
*Mortierella simplex*	CBS 243.82		JX975870	JX976156	[29,30]
*Mortierella stylospora*	CBS 211.32, FSU 9850	Type of *Mortierella stylospora*	JX976086	HQ667359.1	[29,30]
*Umbelopsis isabellina*	CBS 127635		MH864646	MH876082	[32]

## Data Availability

Data are contained within the article and Appendix A.

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
