# Peer review of "Identification and Biological Characteristics of Mortierella alpina Associated with Chinese Flowering Cherry (Cerasus serrulata) Leaf Blight in China"

_jof, 2024, doi:10.3390/jof10010050_

Round 1

Reviewer 1 Report (Previous Reviewer 2)

Comments and Suggestions for Authors

After the previous revision (I used to comment), the quality of the manuscript has been improved. However, I believe it still does not meet the standards yet. Particularly, I’d like give some suggestions as follows:

1.       The English of the current version is still not good enough. I suggest this manuscript being revised by a professional service. Moreover, the connections between sections of the manuscript are not well established, which can be seen below.

2.       In the abstract, what were the label of the 3 selected fungi. Moreover, “The fungus” in line 24 should be the isolates instead. Importantly, the effects and the identification of the three selected isolates have not been shown while the “Identification” term appears in the title.

3.       The paragraph about the disease is not sufficient. Moreover, literatures should be reviewed to increase the persuasiveness of the study. Beside that, other parts of the introduction have been prepared well.

4.       In my opinion, the sample of 30/285 leaves collection is a bit little.

5.       What company is the manufacturer of the PDA medium?

6.       What is ddH2O?

7.       How was pH adjusted?

8.       “designated XA-1 - XA-26” should be rewritten.

9.       Are there any images showing the grow of the fungi on dishes?

10.   The full name should be used in line 203.

11.   The table 2 should be cited with literatures and placed in the materials and methods.

12.   “The morphological traits in this study are 225 consistent with previous research [29]” should be in the discussion not the results.

13.   Why was XA-10 chosen instead of XA-15 and XA-18, while they were all M. alpina? The other two isolates should be investigated as well. This is extremely important. If they are the same isolate with the same ITS, LSU, pathogenicity and morphology, it should be stated.

14.   The whole arguments in lines 296-303 and 330-332 should be cited with literatures.

15.   In line 332, “studies” was used, while only one literature was cited. This is not logical.

16.   The conclusion is too general. It should highlight the findings and significance of the study. The conclusion should be rewritten.

17.   There references are too outdated. There is only one literature from 2023, and Half of the references are before 2018.

Comments on the Quality of English Language

Extensive editing of English language required

Author Response

Reviewer 2 Report (Previous Reviewer 1)

Comments and Suggestions for Authors

I suggest the authors add references in Table 2.

Round 2

Reviewer 1 Report (Previous Reviewer 2)

Comments and Suggestions for Authors

After 2 revisions, the manuscript is almost ready for publication. However, some minor revisions are still required. For example:

1.       In lines 26-27, 3 isolates were stated, then only XA-10 was investigated further. If the isolates are the same, there should have been a statement that the XA-10 isolate was used as a representative, if not, a reason why XA-10 was chosen should be given.

2.       “The 3 isolates resulted in the growth of similar, wavy colonies characterized by dense white aerial mycelia.” This sentence is too confused.

3.       The introduction is good in this current version.

4.       “After screening, 30 leaves were finally selected for pathogen isolation.” Why were 30 out of 285 leaves chosen? Based on what criteria?

5.       “3 isolates (XA-10, XA-15, and XA-18)” I would prefer this to be “the selected isolates”.

6.       “XA-10” should not appear in the material and methods. Just use “selected strain” in this section instead.

7.       The rest of the manuscript has been revised well.

Comments on the Quality of English Language

 Minor editing of English language required

Author Response

This manuscript is a resubmission of an earlier submission. The following is a list of the peer review reports and author responses from that submission.

Round 1

Reviewer 1 Report

Comments and Suggestions for Authors

Revised manuscript: Shao et al._Identification and Biological Characteristics of Mortierella alpina, 2 associated with Chinese flowering cherry (Cerasus serrulata) Leaf 3 Blight in China.

In previous round, I mentioned “In this study, only one isolate XA-10 was obtained and used for further studies. I suggest the authors try to collect more diseased samples and isolates more isolates (strains) to prove the disease mentioned in this study was caused by Motierella alpina.”

I realized that the authors didn’t consider my comments and suggestions.

Comments on the Quality of English Language

No comments.

Reviewer 2 Report

Comments and Suggestions for Authors

1.At first glance, the manuscript contains too many severely grammatical mistakes. Therefore, it should be revised more carefully before submission. Those mistakes could limit the evaluation for the manuscript. Moreover, I recognize that many details which should belong to a certain section appear in another one. For example, materials and methods details appear in the introduction, the result contains both discussion and methods, and the discussion repeat too much result.

2.The symptoms of the disease should be briefly mentioned along with the severity of the disease after the reinoculation.

3.The application of the study is not well informed.

4.The importance of the plant should be more emphasized. In my opinion, the current state of the introduction is not persuasive.

5.The introduction is too fragmented. Some paragraphs share the same aspect, and they should be merged. Please reorganize the introduction.

6.The last part of the introduction should raise the gap and why the study should be conducted along with its objectives. Here, the paragraph shows the materials and methods, which is inappropriate.

7.I believe the XA-10 isolate should be in the result, not the materials and methods. The materials and methods miss the phylogenetic analysis.

8.The result is terribly proposed. There should have been a clear statistical comparison between strains before selecting the XA-10 one. Moreover, the figures do not show any statistical data, so tables should be provided. Furthermore, in the result, citations and method descriptions are not appropriate. The name of the XA-10 strain should be in the Figure 2. The flow of the result is too confused. Particularly, the XA-10 isolate was chosen at the first place because of the Koch’s postulates, but after many tests, the postulates’ results are showed.

9.What is the source of the healthy leaves, and the three other Rosaceae plants?

10.                       There are repeated abbreviations.

11.                       The discussion contains many uncited arguments.

12.                       The conclusion lacks the future application of the study.

Comments on the Quality of English Language

Moderate editing of English language required